# Glucosinolate Profiling and Expression Analysis of Glucosinolate Biosynthesis Genes Differentiate White Mold Resistant and Susceptible Cabbage Lines

**DOI:** 10.3390/ijms19124037

**Published:** 2018-12-13

**Authors:** Md. Abuyusuf, Arif Hasan Khan Robin, Ji-Hee Lee, Hee-Jeong Jung, Hoy-Taek Kim, Jong-In Park, Ill-Sup Nou

**Affiliations:** 1Department of Horticulture, Sunchon National University, 255 Jungang-ro, Suncheon, Jeonnam 57922, Korea; yusuf_agr@pstu.ac.bd (M.A.); gpb21bau@gmail.com (A.H.K.R.); jihee0830@scnu.ac.kr (J.-H.L.); gml79wjd@sunchon.ac.kr (H.-J.J.); htkim@sunchon.ac.kr (H.-T.K.); 2Department of Genetics and Plant Breeding, Bangladesh Agricultural University, Mymensingh 2202, Bangladesh

**Keywords:** white mold, *Sclerotinia sclerotiorum*, glucosinolates, R line, S line, cabbage

## Abstract

Sclerotinia stem rot (white mold), caused by the fungus *Sclerotinia sclerotiorum*, is a serious disease of *Brassica* crops worldwide. Despite considerable progress in investigating plant defense mechanisms against this pathogen, which have revealed the involvement of glucosinolates, the host–pathogen interaction between cabbage (*Brassica oleracea*) and *S. sclerotiorum* has not been fully explored. Here, we investigated glucosinolate profiles and the expression of glucosinolate biosynthesis genes in white-mold-resistant (R) and -susceptible (S) lines of cabbage after infection with *S. sclerotiorum*. The simultaneous rise in the levels of the aliphatic glucosinate glucoiberverin (GIV) and the indolic glucosinate glucobrassicin (GBS) was linked to white mold resistance in cabbage. Principal component analysis showed close association between fungal treatment and cabbage GIV and GBS contents. The correlation analysis showed significant positive associations between GIV content and expression of the glucosinolate biosynthesis genes *ST5b-Bol026202* and *ST5c-Bol030757*, and between GBS content and the expression of the glucosinolate biosynthesis genes *ST5a-Bol026200* and *ST5a-Bol039395*. Our results revealed that *S. sclerotiorum* infection of cabbage induces the expression of glucosinolate biosynthesis genes, altering the content of individual glucosinolates. This relationship between the expression of glucosinolate biosynthesis genes and accumulation of the corresponding glucosinolates and resistance to white mold extends the molecular understanding of glucosinolate-negotiated defense against *S. sclerotiorum* in cabbage.

## 1. Introduction

White mold caused by the fungal pathogen *Sclerotinia sclerotiorum* infects more than 400 plant species, including important crops such as sunflower, chickpea and rapeseed [1,2]. The pathogen usually infects plants as mycelia or airborne ascospores [3]. Although a few fungicides are available to manage this disease, their low efficiencies, the environmental contamination they cause, and the economic costs of both the treatments and the disease itself are substantial [4]. Thus, breeding resistant varieties is the best strategy to control this disease [5]. Although the wide genetic diversity of the members of the Brassicaceae family makes it difficult to draw concrete conclusions, it is generally believed that resistance against *S. sclerotiorum* exists primarily in *Brassica napus* and its relatives [6]. Some efforts have been made to identify resistance resources from wild crucifers, such as *Erucastrum cardaminoides* and *Erucastrum abyssinicum* [7], *Erucastrum gallicum* [8], and *Capsella bursa-pastoris* [9]. More recently, scientists have identified resources with high levels of resistance against *S. sclerotiorum* from wild *Brassica oleracea* [10], one of the parental species of rapeseed. This finding brings new hope for improving *S. sclerotiorum* resistance of rapeseed, especially since its wild relatives, such as *Brassica rupestris*, *Brassica incana*, *Brassica insularis*, and *Brassica villosa*, show high levels of resistance [10]. Completely or highly resistant lines of *B. oleracea* are not currently available. The lack of a resistance source has greatly constrained the breeding of *B. oleracea* for *S. sclerotiorum* resistance, so that little practical benefit has accrued to date from research on *S. sclerotiorum* resistance in *B. oleracea*. Moreover, the interactions between *B. oleracea* and *S. sclerotiorum* leading to eventual pathogenesis have received little attention. Resistance genes and secondary metabolites involved in plant–pathogen interactions provide general resistance to pathogens and insects [11,12,13,14]. In Brassicaceae, glucosinolates (GSLs), vital secondary metabolites biosynthesized from amino acids that are perhaps best known for their anti-oxidative and anti-carcinogenic roles in humans, play important functional roles in the plants’ own resistance to pathogens and insects. GSL metabolism is thus a potentially fruitful source of pathogen-resistance genes.

GSLs can be grouped into three different classes according to the amino acid(s) from which they are derived: aliphatic/alkenyl glucosinolates, derived from methionine; aromatic glucosinolates, derived from phenylalanine and tyrosine; and indole glucosinolates, derived from tryptophan [11]. Among the three classes, aliphatic and indole GSLs are the two most important in Brassicaceae [15,16,17]. GSLs and their hydrolyzed products show significant antimicrobial and insecticidal activities [18] as well as anti-fungal properties in plants [19,20,21,22,23,24,25,26]. GSLs are the precursors of sulfur- and nitrogen-containing secondary metabolites such as isothiocyanates and sulforaphane [27,28,29,30]. The effects of GSL metabolism and sulfur and nitrogen nutrition have been studied because *Brassica* crops contain large amounts of sulfur-containing amino acids and GSLs [31,32,33,34]. In a study on the antimicrobial effect of crude extracts from *Arabidopsis thaliana* [23], 4-methylsulfonyl butyl isothiocyanate was found to be the main active compound, with broad antimicrobial activity, which implied that this GSL-derived isothiocyanate might have a protective effect against plant pathogens. Several investigations revealed alterations of GSL profile upon fungal inoculation [35,36,37,38]. Initial reports described only alteration of indole GSLs [37], but more recent studies have reported alteration of both aliphatic and indole GSLs in response to fungal pathogens [36,38]. Anti-oxidative compounds produced from the degradation of GSLs bring about plants’ defense response against pathogens and herbivores [24,39]. Biotic and abiotic factors, such as pathogen infection, herbivore damage, mechanical injury, and mineral nutrition, can modulate the GSL profile [40,41,42]. Moreover, a wide range of defense reactions can affect GSL content [22,42,43,44].

However, an association between GSL levels and resistance to various fungal pathogens in brassicas has not yet been established [36]. In fact, studies in various *Brassica* species have repeatedly failed to find a strong correlation between pathogen resistance and GSL content following fungal infection, so the overall scenario is perplexing [45,46]. In Arabidopsis, the *MAM1* mutant showed a decrease in GSL level that resulted in susceptibility to *Fusarium oxysporum*, pointing to a protective role of GSLs against fungal infection [23]. GSL levels are also positively correlated with oilseed rape resistance to the pathogen *Sclerotinia sclerotiorum* [47,48], with a few exceptions [43,49,50,51,52]. However, high GSL levels enhance Arabidopsis susceptibility to the fungus *Alternaria brassicicola* [52]. A negative correlation between GSL content in *Brassica napus* and resistance to strains of *A. brassicicola* and *Alternaria brassicae* has also been observed [49]. The inconsistency of the existing data may reflect variations in the behavior of individual fungi (e.g., necrotrophs versus biotrophs) [53], their host specificity (e.g., *Brassica* specialist versus broad spectrum) [54], the genetic purity of the host plants (e.g., isogenic versus heterozygous lines), and the amounts of GSLs and their degradation products produced by the host plant. In cabbage (*B. oleracea*), the relationship between GSL content and resistance to *S. sclerotiorum* has not yet been studied. Several recent reports have shown that resistance to obligate biotrophs, hemibiotrophs, and necrotrophs might be linked to the production of indole GSLs in Brassicaceae [55,56]. Moreover, in *B. oleracea* in particular, the enhanced accumulation of certain aliphatic and indole GSL compounds is associated with concomitant increases in the expression of GSL biosynthesis genes in several inbred lines [35,36,37,57]. These findings on the association of resistance with the GSL profile of *Brassica* species prompted us to examine these plant–pathogen interactions at the molecular and biochemical level. Here, we investigated the correlation between GSL profiles in resistant and susceptible cabbage lines, and the expression of GSL biosynthesis genes upon infection with *S. sclerotiorum*.

## 2. Results

### 2.1. Resistance of Different Cabbage Lines to S. sclerotiorum

We inoculated cabbage leaves of 45 inbred lines of cabbage with *S. sclerotiorum* and observed noteworthy differences in response in terms of the appearance of disease symptoms. In particular, one line, SCNU-C-049 (denoted hereafter as the resistant or R line), exhibited complete resistance with no disease symptoms, while the other 44 lines were susceptible to *S. sclerotiorum*, as evidenced by changes visible at 5 days post inoculation (DPI) (Appendix A). We confirmed the resistance of SCNU-C-049 and the susceptibility of one selected S line (SCNU-C-033) through three repeated experiments, in which infected leaf, stem and head of both the R line and the selected S line showed consistent disease reactions (Figure 1). 

### 2.2. Overview of Individual GSL Profiles in Cabbage Lines

Another study conducted by our group had previously produced a sample illustrative spectrum by mass spectrometry analysis of samples from cabbage lines that were resistant and susceptible to a different pathogen, ringspot (*Mycosphaerella brassicicola*), and used it to identify individual glucosinolate compounds found in cabbage leaves [38]. We performed high-performance liquid chromatography (HPLC) analysis to detect eleven GSL compounds identified in that earlier work in our R (SCNU-C-049) and S (SCNU-C-033) cabbage lines: glucoiberin (GIB), progoitrin (PRO), glucoraphanin (GRA), sinigrin (SIN), glucoerucin (GER), gluconapin (GNA), glucoiberverin (GIV), hydroxyglucobrassicin (HGBS), glucobrassicin (GBS), methoxyglucobrassicin (MGBS), and neoglucobrassicin (NGBS) (see HPLC peaks and GSL contents in Appendix A and Appendix A). In non-inoculated control plants, the amounts of both aliphatic and indole GSLs (i.e., PRO, GRA, MGBS, and GBS) differed significantly from those in the inoculated R and S plants (Figure 2). Inoculating cabbage plants with *S. sclerotiorum* significantly changed the GSL profiles in the leaves of both the R and S lines. In the R line, the level of GIV was significantly higher, by 1.46-fold, in inoculated plants at 3 DPI, compared to that in mock-treated plants (Figure 2). In contrast, the GIV level was not significantly altered after fungal infection in the S line (Figure 2, Appendix A). Similarly, the level of GBS in the R line was 5.28-fold higher in the treated compared to the mock-treated plants at 3 DPI (Figure 2, Appendix A), whereas the S line did not show significant changes in GBS level after infection. Thus, GIV and GBS showed an increasing trend after infection only in the R line. 

In the S line, the levels of GNA were significantly higher, by 2.34-fold, at 3 DPI in the treated compared to the mock-treated plants, whereas the GNA level was not significantly altered in the R line after infection (Figure 2, Appendix A). Overall, our results showed that the contents of both aliphatic GIV and indole GBS increased in the R line, but not the S line, starting at the time of infection (Figure 2). In addition, the total GSL content did not vary significantly in either the R or the S line at 3 DPI (Appendix A).

### 2.3. Upregulation of MYB28-Bol017019, MYB34-Bol017062, ST5a-Bol026200, ST5a-Bol039395, ST5b-Bol026202 and ST5c-Bol030757 in the R Line after Inoculation

In order to investigate whether the expression levels of genes belonging to the aliphatic and indole GSL biosynthesis pathways, and their associated transcription factors, are associated with plant resistance upon *S. sclerotionum* infection, reverse transcription quantitative PCR (RT-qPCR) was performed. The *Actin* gene (*actin1*) was used for normalization of expression of 38 target genes. The expression levels and melting curves of the *Actin* gene and 38 glucosinolate-biosynthesis-related genes are given in Appendix A and Appendix A, respectively. The forward and reverse primers of *Actin1* were designed based on the sequence available at NCBI database (GenBankAccession no. AF044573). The selection of that gene as reference gene to perform expression data normalization was based on previous reports showing the stability of that gene in the same plant species upon similar experimental conditions [35,58]. Here we report the changes in the transcription levels of genes encoding aliphatic and indole GSL biosynthesis and their associated transcription factors that resulted from *S. sclerotiorum* infection. We found that two genes encoding transcription factors—one affecting the synthesis of an aliphatic GSL, *MYB28-Bol017019*, and one that of an indole GSL, *MYB34-Bol017062*—showed 10.21-fold and 3.08-fold upregulation at 1 DPI in the R line after infection, respectively, as compared to their expression in mock-treated plants (Figure 3, Table 1, and Appendix A). Two aliphatic biosynthesis genes, *ST5b-Bol026202* and *ST5c-Bol030757*, and two indole biosynthesis genes, *ST5a-Bol026200* and *ST5a-Bol039395*, were also significantly upregulated in the R line at 3 DPI: *ST5b-Bol026202* and *ST5c-Bol030757* exhibited 5.6-fold and 68.59-fold upregulation, whereas *ST5a-Bol026200* and *ST5a-Bol039395* exhibited 82.90-fold and 25.30-fold upregulation, respectively (Figure 3, Table 1, and Appendix A). 

### 2.4. Upregulation of Transcription Factor-Related Genes and GSL Biosynthesis Genes in the S Line after Inoculation

We measured the expression levels of 10 GSL biosynthesis genes in the control, mock-treated, and infected plants. In the uninfected control plants, one indole GSL biosynthesis gene, *CYP81F1-Bol017376*, showed significantly higher expression in the R line compared to the S line (Figure 4). In the S line at 1 DPI, *MYB28-Bol036743*, *MYB29-Bol008849*, *MYB28-Bol036286*, and *MYB28-Bol007795*, which encode transcription factors genes related to the aliphatic GSLs, showed increased expression (of 5.8-, 51.0-, 21.6-, and 2.9-fold, respectively) in the treated compared to the mock-treated plants (Figure 4, Table 1, Appendix A). In addition, the aliphatic GSL biosynthesis genes *FMOGS-OX2-Bol010993*, *AOP2-Bo3g052110*, *AOP2-Bo9g006240*, and *GSL-OH-Bol033373* had 5.5-, 8.4-, 11.4-, and 65.3-fold higher expression, respectively (Figure 4, Table 1, Appendix A). Among the indole GSL biosynthesis genes, *CYP81F1-Bol017375* had 2.0-fold higher expression at 1 DPI and *CYP81F1-Bol017376* had 7.96-fold higher expression at 3 DPI in the S line in the treated compared to the mock-treated plants (Figure 4, Table 1 and Appendix A).

### 2.5. Upregulation of Transcription-Factor-Related Genes and GSL Biosynthesis Genes in Both R and S Lines

Among the 38 transcription-factor-related genes and GSL biosynthesis genes investigated, the expression of 10 genes was upregulated in both R and S lines after inoculation, compared to that in mock-treated plants (Table 1, Appendix A). *MYMYB122-Bol026204*, which encodes a transcription factor related to indole GSLs, had 189-fold increased expression in the R line at 3 DPI (Table 1, Appendix A, and Appendix A). Increased expression was also found in infected plants of the R line at 3 DPI for the following indole GSL biosynthesis pathway genes: *CYP81F4-Bol032712* by 21.1-fold, *CYP81F2-Bol026044* by 54.5-fold, *CYP81F2-Bol014239* by 871-fold, *CYP81F2-Bol012237* by 48.3-fold, *IGMT1-Bol007029* by 24.7-fold, and *IGMT2-Bol007030* by 171-fold (Table 1, Appendix A, Appendix A). In contrast, the S line showed much lower upregulation of these genes after infection at 3 DPI compared to that in the mock-treated plants: *CYP81F4-Bol032712* expression was increased by 14.5-fold, *CYP81F2-Bol026044* by 44.9-fold, *CYP81F2-Bol014239* by 1135-fold (1.75-fold lower than in the R-line), *CYP81F2-Bol012237* by 12.5-fold, *IGMT1-Bol007029* by 10.3-fold, and *IGMT2-Bol007030* by 41.1-fold (Table 1, Appendix A, Appendix A). Meanwhile, we observed inconsistent responses in the expression of 12 genes (*MYB34-Bol007760, MYB34-Bol036262, MYB51-Bol013207, MYB51-Bol030761, ST5b-Bol026201, CYP81F1-Bol028913, CYP81F3-Bol032711, CYP81F3-Bol028919, CYP81F4-Bol028918, FMOGS-OX5-Bol029100, FMOGS-OX5-Bol031350*, and *AOP2-Bo2g102190*) in response to fungal treatments in the R and S lines (Appendix A).

### 2.6. Correlation between the Levels of Individual GSLs and the Expression Level of GSL Biosynthesis Pathway Genes Induced by S. sclerotiorum in the R and S Lines 

Heat maps of the fold changes in the expression levels of transcription-factor-related genes and GSL biosynthesis genes that we observed after pathogen inoculation emphasized that these changes were consistent with the changes in the levels of individual GSLs measured in the R and S lines after infection, as compared to the mock-treated controls (Figure 5). We obtained the highest significant positive correlation values for Pearson’s correlation coefficient between the levels of the aliphatic GSLs GIB and SIN and the expression of *ST5b-Bol026201*, between the PRO and GRA levels and *AOP2-Bo2g102190* expression, and between the GIV level and *ST5b-Bol026202* and *ST5c-Bol030757* expression. In contrast, no significant positive correlation was found for GNA and GER (Figure 5A, Appendix A). Among indole GSLs, the GBS level showed significant positive correlation with *MYB122-Bol026204*, *ST5a-Bol026200*, *ST5a-Bol039395*, *IGMT1-Bol007029,* and *IGMT2-Bol007030* expression, the HGBS level with *MYB34-Bol036262* and *CYP81F3-Bol028919* expression, and the MGBS level with *MYB34-Bol036262, MYB122-Bol026204*, *ST5a-Bol026200*, *ST5a-Bol039395,* and *IGMT2-Bol007030* expression, whereas the NGBS level had the highest significant positive correlation with *MYB34-Bol007760* expression (Figure 5B, Appendix A). 

Principal component analysis (PCA) for the contents of the 11 individual GSL compounds under five different treatment combinations in the R line SCNU-C-049 and the S line SCNU-C-033 of cabbage that we tested revealed an association between white mold resistance and the pattern of GSL accumulation. There were major contrasts among the contents of the individual GSLs. The first four PCs explained 88.3% of the total variation in the datasets (Appendix A). PC1 and PC2 accounted for 43.5% and 24.2% of the total variation, respectively, largely corresponding to higher positive coefficients versus lower negative coefficients of individual GSL profiles (Appendix A). PC1 clearly distinguished the R line from the S line, based on individual GSL profiles (Figure 6 and Appendix A) for their positive and negative coefficients respectively. PC2, on the other hand, showed positive association with GIV and GBS in infected samples at 3 DPI (T3) for their positive coefficients, compared to the mock-treated and control samples for their negative coefficients (Figure 6).

## 3. Discussion

### 3.1. Resistance of Cabbage Lines to S. sclerotiorum

We screened 45 inbred lines of cabbage (*B. oleracea* var. *capitata*) and found one completely resistant line, SCNU-C-049 (Figure 1 and Appendix A), whereas a previous report had shown high levels of resistance against *S. sclerotiorum* in wild *B. oleracea* [10]. The genotypes screened in this study were developed through breeding. Thus, the observed differences explain genotypic variation and indicate that resistance towards *S. sclerotiorum* is genotype specific. Therefore, the factors controlling the resistance could be transferred to elite cabbage lines.

### 3.2. Levels of Total GSLs, GIV, and GBS Were Related to White Mold Resistance

This study revealed a clear association between plant resistance and GSL accumulation due to pathogen inoculation. In both the R and S lines, the total GSL content, as measured at 3 DPI, did not change significantly after *S. sclerotiorium* inoculation, which indicated that total GSL was not correlated to either resistance or susceptibility in cabbage (Appendix A). These findings were consistent with the results of other studies indicating that pathogen resistance in different *Brassica* species is not strongly correlated with the overall level of GSLs in each species [45,46]. Notably, we found that the levels of many individual GSLs were altered in the mock-treated plants compared to the control plants. For example, the accumulation of GIB, PRO, GRA, SIN GER, GNA, GBS, HGBS, GBS, and NGBS was generally variable in both control and mock-treated plants of both R and S lines (Figure 2). Because of this variability in GSL accumulation in the absence of infection, we compared changes in the contents of the individual GSLs in the samples from the R and S plants after infection to those in mock-treated samples, as a reference. The contents of aliphatic GIV and indole GBS increased in the R line after inoculation (Figure 2), indicating that white mold resistance in cabbage may be accomplished through the accumulation of both aliphatic and indole GSLs. These results were consistent with those of some past studies [36], but not others [53,56,59,60]. In general, however, these data suggest that increased levels of aliphatic GIV and indole GBS may confer resistance to *S. sclerotiorum* in cabbage.

### 3.3. Increased Expression of ST5b-Bol026202 and ST5c-Bol030757 Led to Increased GIV in the R Line

Secondary alterations of the desulfoglucosinolates GIB and GIV, and other aliphatic GSLs, are linked with the *ST5b* and *ST5c* genes (Appendix A). In this study, increased expression of *ST5b-Bol026202* and *ST5c-Bol030757* was associated with higher levels of GIV biosynthesis (Figure 5A, Appendix A), confirming results from a previous study [35,38]. Therefore, our data imply that infection-induced upregulation of these genes leads to an increase in the level of GIV, which is linked with white mold resistance (Figure 2 and Figure 3), although molecular studies will be needed to further validate the associations based on these correlations.

### 3.4. Increased Levels of Aliphatic GIV and Indole GBS Were Associated with White Mold Resistance

The pathogen-induced accumulation of GIV seen in the R line, in contrast with the quite static accumulation of GIV in the S line, indicated that GIV has an important role in the resistance conferred by GSL accumulation (Figure 2). In addition, the increased levels of the indole GBS observed in the R as compared to the S line also likely contribute to white mold resistance (Figure 2). These results were consistent with previous observations of higher accumulation of GIV, GBS, and MGBS associated with ringspot resistance in cabbage [38], and increased accumulation of GIV, GBS, and NGBS associated with blackleg resistance in cabbage [37]. 

### 3.5. Expression of MYB28 and MYB34 Likely Induced Expression of GSL Biosynthesis Genes, Leading to Increases in GBS in the R Line

In Arabidopsis, biotic challenges are believed to be regulated by the upregulation of *MYB28*, a novel regulator of aliphatic glucosinolate biosynthesis genes [61]. *MYB34* genes directly control the biosynthesis of indole GSLs in Arabidopsis [62] and *B. oleracea* [35,57]. Moreover, in Arabidopsis, *MYB34*, in conjunction with *MYB51* and *MYB122*, takes part in resistance against *Plectosphaerella cucumerina*, where the indole-GSL-breakdown-related gene *PENETRATION2* (*PEN2*) plays a major role in triggering the expression of relevant biosynthesis genes upon pathogen inoculation [59]. A *MYB34* (*Bol007760*) gene is also induced in response to mimic biotic elicitation with methyl jasmonate (MeJA) in broccoli (*Brassica oleracea* var. *italica*), indicating that this response might follow jasmonic acid (JA) signaling [57]. In our study, the expressions of *MYB28-Bol017019* and *MYB34-Bol017062* increased by 10.2- and 3.08-fold, respectively, at 1 DPI in R plants inoculated with *S. sclerotiorum* compared to the mock-treated plants (Figure 3). *MYB28-Bol017019* and *MYB34-Bol017062* may play roles in the transactivation of genes required for the biosynthesis of indole GSLs in response to *S. sclerotiorum* infection (Figure 2). In our results, the upregulation of *MYB28-Bol017019* and *MYB34-Bol017062* expression in the R line was associated with the accumulation of aliphatic GIV and indole GBS, respectively, an observation that is also supported by earlier results [35]. In contrast, a number of genes were highly expressed only in the S line after inoculation, indicating that these genes are either not the key regulators, or that any glucosinolate altered by them (for example, gluconapin) has no important role in defense against *S. sclerotiorum* infection (Table 1, Figure 4). From a recent investigation in *B. oleracea*, it is evident that lower accumulation of a GSL compound is not always consistently related to lower expression of one or a few specific GSL biosynthesis genes in all genotypes, and vice versa [35].

### 3.6. Accumulation of Indole GBS in the R Line Was Activated by Increased Expression of GSL Biosynthesis Genes

GBS, which plays a role in antifungal responses in plants, showed increased abundance in the R line, as compared to that in mock-treated plants, in response to *S. sclerotiorum* infection, and this was associated with a significant upregulation of the expression of *CYP81F4-Bol032712*, *CYP81F2-Bol026044*, *CYP81F2-Bol014239*, and *CYP81F2-Bol012237* (Appendix A). In a previous study, MeJA treatment increased the expression of *CYP81F4* by 2400-fold in broccoli and 10-fold in cabbage [57], suggesting that resistance against *S. sclerotiorum* is governed by signaling pathways involved in the metabolism of indole GSLs. Experimental evidence has suggested that resistance against necrotrophic pathogens could be modulated by the jasmonic acid/ethylene (JA/ET) signaling pathway [63,64,65,66]. Since *S. sclerotiorum* is a necrotrophic fungus, it is therefore likely that either the JA or ET signaling pathway might be involved in the resistance response of cabbage lines [67,68]. Another study also found a similar association between *CYP81F2* expression and GBS levels in *B. oleracea* [69]. The accumulation of GBS and the changes in the expression of *CYP81F1-Bol028914, CYP81F2-Bol012237, CYP81F2-Bol014239, CYP81F2-Bol026044, CYP81F4-Bol032712, CYP81F4-Bol032714, IGMT1-Bol007029, IGMT1-Bol020663*, and *IGMT2-Bol007030* by 0.12-, 2.31-, 1.75-, 2.18-, 1.72-, 0.74-, 5.72-, 2.27-, and 6.30-fold, respectively, in the R as compared to the S line at 3 DPI (Appendix A) indicate that the accumulation of particular GSL components may be associated with physiological responses mediated by gene functions.

The quantities of GSLs in leaf tissues are the result of simultaneous activation of myrosinases (biosynthesis and catabolism), which can upregulate the abundance of GSL components at a specific time period. In vitro studies have reported that MGBS, as well as SIN and GBS [70], has antifungal activity, and that increased accumulation of MGBS confers moderate resistance to *Leptosphaeria maculans* in cabbage plants [36]. Here, we observed an increase in GBS level in the R line compared to the S line after *S. sclerotiorum* infection, along with upregulation of *CYP81F4-Bol032712*, *CYP81F2-Bol026044*, *CYP81F2-Bol014239*, and *CYP81F2-Bol012237*, which encode proteins involved in methoxylation and the conversion of GBS to 4-MGBS (Figure 2). It is likely that both GSL content and GSL biosynthesis pathway genes serve to confer resistance to *S. sclerotiorum*. Our findings also agree with the report that in Arabidopsis, upon infection with *Blumeria graminis Erysiphe pisi* and *Plectosphaerella cucumerina*, *CYP81F2* expression induces antifungal defenses [55].

### 3.7. Accumulation of Aliphatic GIV with Expression of ST5b-Bol026202 and ST5c-Bol030757 and Indole GBS with ST5a-Bol026200 and ST5a-Bol039395 May Play a Role in Resistance 

A notably consistent association between GSL content and expression levels of genes was observed in our heat map (Figure 5). The Pearson’s correlation coefficient showed the highest significant positive correlations between the levels of the aliphatic GIV and the expressions of *ST5b-Bol026202* and *ST5c-Bol030757*, and between the levels of the indolic GBS and the expressions of *ST5a-Bol026200* and *ST5a-Bol039395* (Figure 2, Figure 3 and Figure 5). These data show that changes in the expression levels of these genes correlate to the contents of individual GSLs in response to *S. sclerotiorum* infection. The PCA showed a strong association between the accumulation of GIV and GBS in the R line at 3 DPI for their positive coefficients at PC1 and PC2 (Figure 6). These GSLs might function in the regulation of resistance to white mold in cabbage. These results were also supported by a previous observation that the contents of aliphatic and indole GSLs were correlated with complete resistance to blackleg in cabbage [36]. Lastly, the upregulation of GSL biosynthesis genes occurred within 1 to 3 days after inoculation, at the time of the first appearance of disease symptoms. GSLs began to accumulate at 3 DPI. This suggests that at the time when symptoms first appeared, GSL biosynthesis genes were induced in order to initiate a GSL-mediated resistance response.

### 3.8. Association of GSL Biosynthesis Genes and Accumulation of Individual GSLs in the S Line

Ten GSL biosynthesis genes were highly expressed in the S line at 1 DPI (Figure 4). The PCA analysis showed positive association among individual GSLs and the S line (i.e., GIB, SIN, and GNA) in regard to their negative coefficients across PC1 (Figure 6). Among these individual GSLs, only one aliphatic GSL, GNA, accumulated significantly in the S line (Figure 2). Therefore, GNA may confer susceptibility to *S. sclerotiorum* in cabbage, which is consistent with results from another study indicating that GNA exhibits increased accumulation in clubroot-susceptible *B. napus* plants and is likely a key factor in the pathogenesis of clubroot disease [71]. 

## 4. Materials and Methods

### 4.1. Plant Materials and Growth Conditions

Seeds of 45 cabbage (*B. oleracea* var. *capitata*) inbred lines (Appendix A) were germinated in multi-pot trays using coco-peat soil in a growth chamber at 24 °C and 60% relative humidity (RH), with a 16 h/8 h (light/dark) photoperiod. When the seedlings bore two visible leaves (10 × 10 × 12 cm), they were transplanted to large pots, one plant per pot, filled with a mixture of 50% coco-peat and 50% soil. Plants were inoculated at the ninth-leaf stage (third leaf, stem) and were kept in an inoculation chamber (24 °C, 98% RH) covered with black polyvinyl to maintain high RH, as it positively influences disease progression [72]. They were evaluated for white mold disease one month after head formation. 

### 4.2. Inoculum Preparation

An isolate (Muan) of *S. sclerotiorum*, obtained originally from field-sown cabbage plants in Muan, South Korea, was maintained and cultured on potato dextrose agar (PDA; 25% potato, 2.5% dextrose and 1.5% agar, pH 5.8). The isolate was maintained at 4 °C in darkness, and cultured twice before inoculation at 23 °C in darkness. Mycelial agar plugs (7 mm in diameter) punched from the margin of a 3-day-old culture of *S. sclerotiorum* grown on PDA were used as the inoculum [73,74].

### 4.3. Inoculation Technique and Disease Assessment

We reared 15 seedlings against each genotype to obtain seedlings of homogenized growth at the day of inoculation. We used those plants that reached ninth leaf at the day of inoculation. Three different plants from each of the 45 inbred lines (biological replicates) were inoculated in the third-youngest leaf of ninth-leaf-stage plants. Control plants remained undisturbed, whereas mock-treatment was done with mycelium-free agar plugs. Two inoculation procedures described in Zhao and Meng [74,75] and Yu et al. [67] were used, with modifications (Appendix A), to assess resistance to *S. sclerotiorum*. The first procedure was inoculation of the third leaf of ninth-leaf-stage plants with mycelial agar plugs in a growth chamber, to evaluate leaf resistance. The mycelial agar plug was inoculated on the middle of each leaf. The inoculated leaves were sprayed with a fine mist of water and covered with black polyvinyl to maintain a high level of relative humidity, and the plants were kept at 24 °C in darkness. The lesion was measured from 1 to 5 days post inoculation (DPI) in all lines, and was standardized with presence and absence of disease symptoms. The second procedure was stem inoculation with mycelial agar plugs that was conducted in a different set of plants, to measure stem resistance at the same stage. Stems of each inbred line in each replicate were inoculated with mycelial agar plugs at a height of 20 cm above the ground. Each plug was affixed with alpin and plastic wrap to ensure close contact of the inocula with the stem surface, and to maintain humidity. The plants were sprayed with water mist every day after inoculation for 3 days. The stem lesions were measured at 7 DPI. In addition to using these two procedures, we also inoculated cabbage heads (one month after head formation) and then observed the head lesions at 10 DPI (with the head right side up) and at 15 DPI (with the head upside down). Leaf disease progression was examined in the selected R line (SCNU-C-049) and S line (SCNU-C-033) up to 5 days post inoculation (DPI) to make a decision on the sampling sites for RNA extraction and HPLC analysis (Figure 7).

### 4.4. Leaf Sampling and Preparation for HPLC and Gene Expression Analysis

Leaf samples at 1 and 3 DPI were simultaneously sampled from the R line and the S line from each of the control, mock-treated, and *S. sclerotiorum*–infected plants, both to evaluate the levels of endogenous GSLs and to quantify the expression of GSL biosynthetic pathway genes (Appendix A). Inoculated but non-infected and mock-treated leaf pieces were collected at sampling. The collected samples were flash-frozen in liquid nitrogen and immediately stored in a −80 °C freezer until needed for reverse transcription quantitative PCR (RT-qPCR) and HPLC analyses.

### 4.5. GSL Content Measurements

Desulfoglucosinolates were extracted from leaf samples from three biological replicates for each of the control, mock-treated, and *S. sclerotiorum*–infected plants, via a modified HPLC protocol as previously described [35,57,76]. The leaf samples stored at −80 °C were ground to a very fine powder after methanol treatment. The ground-up leaf tissues were kept at 70 °C for 10 min, then at room temperature for about an hour, and then centrifuged for 8 min at 10,000× *g* at 4 °C to remove undesirable sediment. The supernatant was subjected to anion-exchange chromatography, and the resulting effluent was considered the crude GSL sample. This crude GSL sample was then desulfurized as previously described [35,57,76], and then passed through an elution process with 1 ml of distilled water. The samples were subjected to high-speed centrifugation at 20,000× *g* for 4 min at 4 °C, and then filtered using a polytetrafluoroethylene filter (13 mm, 0.2 μm; Advantec, Pleasanton, CA, USA). Purified GSLs were analyzed by HPLC on a Waters 2695 HPLC system (Waters, Milford, MA, USA) equipped with a C18 column (Zorbax Eclipse XBD C18, 4.6 mm × 150 mm; Agilent Technologies, Palo Alto, CA, USA). Water and acetonitrile were used as the mobile-phase solvents. The content of individual glucosinolates was measured at a wavelength of 229 nm using a PDA996 UV-visible detector (Waters). A standard curve was used for the quantification of the identified GSLs, with sinigrin (SIN) as a standard. HPLC-MS analysis (Agilent 1200 series, Agilent Technologies) was used for the identification of individual GSLs [57].

### 4.6. Primer Design for Expression Analysis of GSL Biosynthesis Genes

We selected 38 genes encoding proteins involved in GSL biosynthesis pathways, of which 11 are transcription factors: five related to the aliphatic and six to the indole biosynthesis pathways. Of the remaining 27 genes, 10 encode proteins involved in aliphatic GSL biosynthesis, and 17 proteins involved in indole GSL biosynthesis (Appendix A, Appendix A) [35,57]. The primers were previously designed, and their efficiencies were calculated following Robin et al. [35]. In order to test the efficiency of primers, pooled cDNA of cabbage inbred lines of the same concentration of 300 ng·µL^−1^ was serially diluted 10 times per dilution. One µL samples of diluted cDNA at 10^0^, 10^−1^, 10^−2^, 10^−3^, 10^−4^ and 10^−5^ concentrations of original were used as templates in each reaction, with forward and reverse primers. The qRT-PCR reaction was conducted for 40 cycles with melting curves in triplicate without template control. Semi-log plots of Ct versus fold dilutions were drawn to determine the slope. Primer efficiencies were calculated using the following equation: e = 10^^(−1/slope)^. A value of slope of −3.321928 indicated 100% efficiency of a primer. Primers with efficiency levels of 90% to 100% were selected for expression analysis. Primers were often redesigned to obtain efficiency within the expected range.

### 4.7. cDNA Synthesis and RT-qPCR Analysis

Total RNA was extracted from frozen, cold-treated leaf samples of cabbage with an RNeasy Mini kit (Qiagen, Hilden, Germany), following the manufacturer’s guidelines. DNA was removed from the samples using RNase-free DNase (Promega, Madison, WI, USA), also according to the manufacturer’s instructions. Purity of the extracted RNA was determined by the 260/280 nm ratio as quantified with a Nanodrop^®^ ND-1000 and NanoDrop v3.7 software (Thermo Fisher Scientific, Waltham, MA, USA) (Appendix A). The integrity of RNA was checked by agarose gel electrophoresis [77]. Complementary DNA (cDNA) was synthesized by using 5 µg of extracted RNA and an oligo (dT) primer of first strand cDNA synthesis kit (Invitrogen, Madison, WI, USA) following the manufacturer’s instructions, and the equality of cDNA for the samples was normalized by comparing the thickness of the PCR amplicons from the *B. oleracea actin1* gene [58]. RT-PCR was conducted using 1 μL of cDNA, 1 μL of forward and reverse primers (10 pmol concentration), 8 μL of Prime Taq Premix (2×) containing 1 U Taq polymerase (GENETBIO Inc., Korea), and 9 μL of double distilled water, with a total volume of 20 µL. PCR conditions: 5 min initial denaturation followed by denaturation at 94 °C and 25 cycles of denaturation at 94 °C for 30 s, annealing at 56 °C for 30 s, extension at 72 °C for 45 s, and final extension at 72 °C for 7 min. PCR products were run on a 1.5% agarose gel along with a 100 bp size DNA ladder, stained with HiQ blue mango (20,000×), and pictured under UV light to obtain expected amplicons. A quantitative PCR (qPCR) was performed using iTaqTM SYBRR Green Super-mix with ROX dye (Bio-Rad, Hercules, CA, USA) to investigate the expression levels of GSL biosynthesis genes. Each reaction was carried out in a 20 μL total volume containing 1 μL cDNA template (60 ng μL^−1^ concentration), 1 μL forward and 1 μL reverse primers (both of 10 pmol concentration), 10 μL iTaqTM SYBRR Green Super-mix (Bio-Rad, Hercules, CA, USA), and 7 μL ultra-pure water. The qPCR was run to conduct denaturation, annealing, and amplification, with the following set conditions: 95 °C for 10 min followed by 40 cycles of 95 °C for 20 s, 58 °C for 20 s, and 72 °C for 30 s. Signal acquisition was performed for each sample, and the fluorescence intensity was recorded at the end of each cycle. Individual biological samples were read three times as technical replicates. The quantification cycle (Cq) analysis was conducted using Light Cycler 96 software (Roche, Mannheim, Germany), and a popular method, the Livak’s comparative 2^−^^△△^*^C^*^t^ method [78], was used for calculating the relative expression of each gene. 

### 4.8. Statistical Analysis

One-way analysis of variance (ANOVA) was performed to test the statistical significance of the differences in the results of the different treatments between the R line, SCNU-C-049, and the S line, SCNU-C-033, using Minitab 18 statistical software (Minitab Inc., State College, PA, USA). A heat map was drawn in Microsoft Excel to show the correlation between GSL content and GSL biosynthetic gene expression according to each specific treatment of the R and S lines, using conditional formatting options (Appendix A). To explore statistical significances of the differences among the treatments, a one-way ANOVA was conducted, followed by Tukey’s pairwise comparison as a post hoc test. Relevant statistical measures, including ANOVA, for individual GSL contents and expression level of genes are presented in Appendix A. A principal component analysis with correlation matrix command was conducted using Minitab 18 (State College, PA, USA) statistical software to explore association between glucosinolate contents after *S. sclerotiorum* inoculation and resistance of cabbage lines. 

## 5. Conclusions

The GSL profiling and expression analysis of GSL-related genes in cabbage infected by *S. sclerotiorum* identified a direct association between the expression of the genes and the abundances of the corresponding GSLs in a resistant and a susceptible line of cabbage. This study showed that the simultaneous accumulation of pathogen-induced aliphatic GIV and indole GBS were associated with white mold resistance in cabbage. Noteworthy differences in expression at 3 days post inoculation were observed for two genes encoding aliphatic biosynthetic proteins, *ST5b-Bol026202* and *ST5c-Bol030757*, and two genes encoding indole biosynthesis genes, *ST5a-Bol026200* and *ST5a-Bol039395*. The GSLs and the corresponding genes identified in this study are candidate genetic and biochemical determinants of resistance, and could be tested in efforts to improve white mold resistance in cabbage.

## Figures and Tables

**Figure 1 ijms-19-04037-f001:**
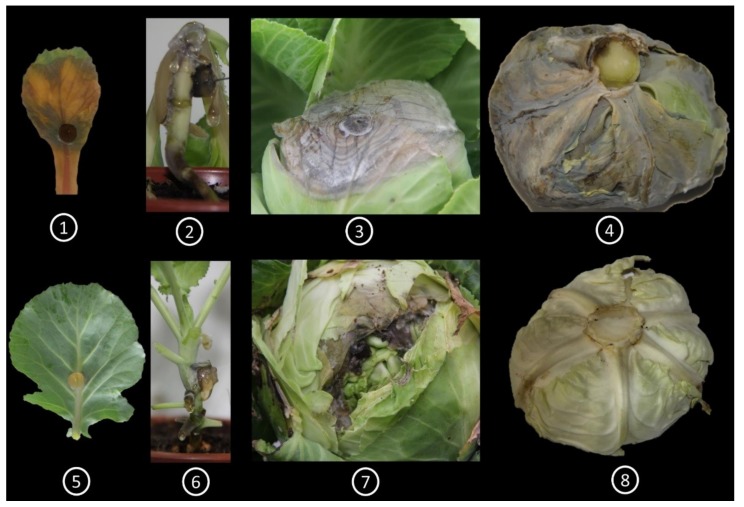
Leaf, stem and head bioassay of the susceptible line SCNU-C-033 (**1**, leaf; **2**, stem; **3**, head seen from top; **4**, head seen from bottom) and resistant line SCNU-C-049 (**5**, leaf; **6**, stem; **7**, head seen from top; **8**, head seen from bottom) of cabbage inoculated with *S. sclerotiorum*. Photographs were taken on the following days post inoculation (DPI): leaf, 5 DPI; stem, 7 DPI; head from top, 10 DPI; head from bottom, 15 DPI.

**Figure 2 ijms-19-04037-f002:**
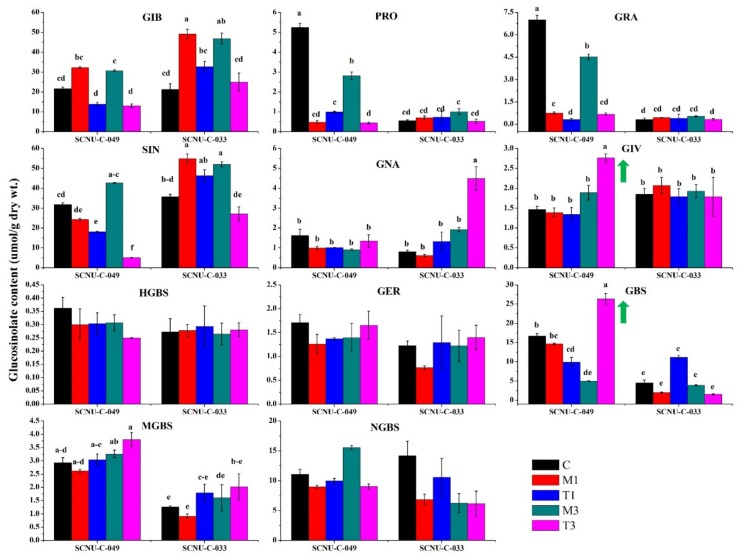
Contents of 11 individual glucosinolates in leaf samples from R (SCNU-C-049) and S (SCNU-C-033) lines of cabbage under different treatment conditions (C, control; M1, mock day 1; T1, treated day 1; M3, mock day 3; T3, treated day 3). The means of three biological replicates are presented. Vertical bars indicate standard deviation. Different letters indicate statistically significant differences between R and S lines and treatment interactions. Upward-pointing green arrows indicate increased glucosinolate content of R line in response to *S. sclerotiorum* infection. R, resistant; S, susceptible. HPLC–mass spectrometry (HPLC-MS) analysis (using an Agilent 1200 series instrument, Agilent Technologies) was conducted following Abuyusuf et al. [38].

**Figure 3 ijms-19-04037-f003:**
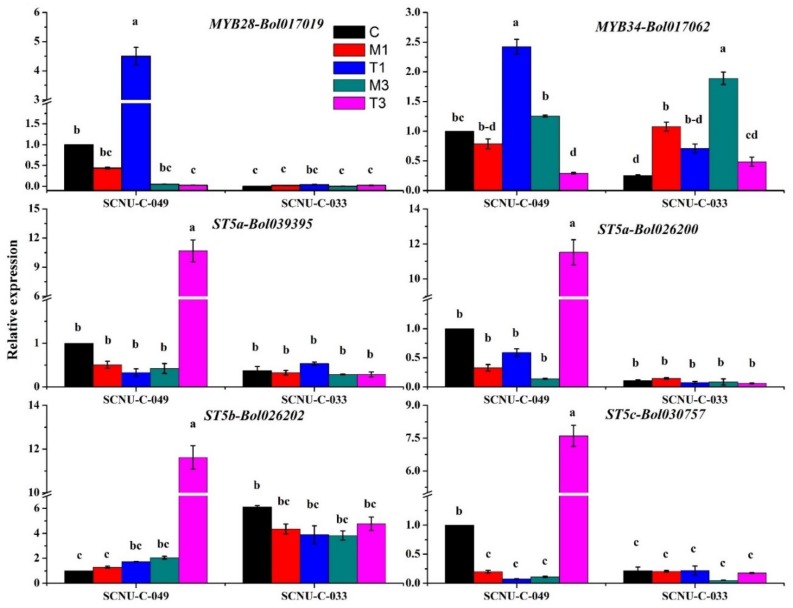
Upregulation of *MYB28-Bol017019, MYB34-Bol017062, ST5a-Bol026200, ST5a-Bol039395, ST5b-Bol026202* and *ST5c-Bol030757* genes in white-mold-inoculated R (SCNU-C-049) line at 1 and 3 DPI compared to mock-treated plants. No similar upregulation was seen in a susceptible (S; SCNU-C-033) line. C, control; M1, mock day 1; T1, treated day 1; M3, mock day 3; T3, treated day 3. The means of three biological replicates are presented. Vertical bars indicate standard deviation. Different letters indicate statistically significant differences between R and S lines and treatment interactions. R, resistant; S, susceptible.

**Figure 4 ijms-19-04037-f004:**
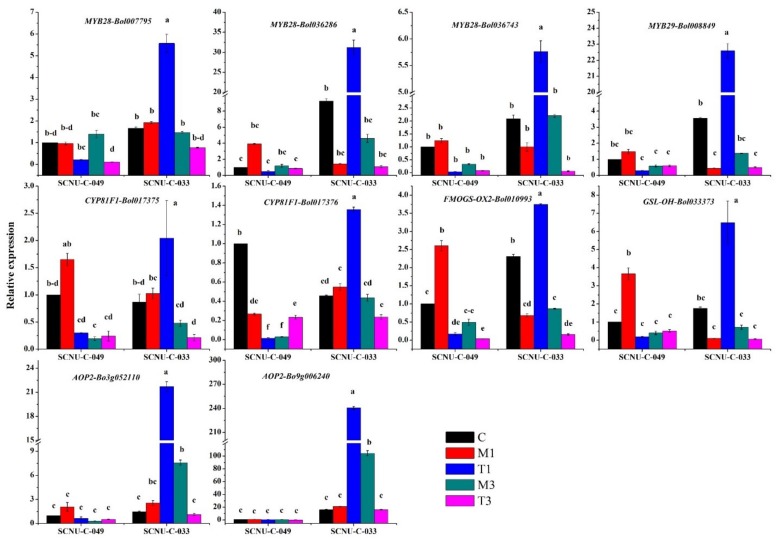
Upregulated transcription-factor- and glucosinolate-biosynthesis-related genes in white-mold-inoculated S (SCNU-C-033) cabbage at 1 DPI compared to mock-treated plants. No similar upregulation was seen in a resistant (R; SCNU-C-049) line. C, control; M1, M3, mock day 1; T1, treated day 1; M3, mock day 3; T3, treated day 3. The means of three biological replicates are presented. Vertical bars indicate standard deviation. Different letters indicate statistically significant differences between R and S lines and treatment interactions. R, resistant; S, susceptible.

**Figure 5 ijms-19-04037-f005:**
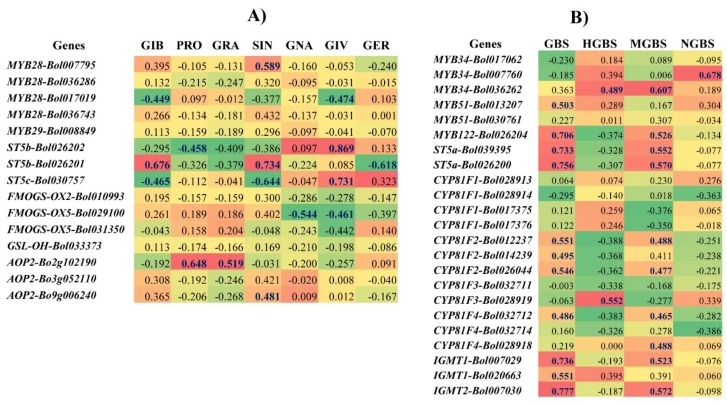
Heat maps showing correlation between the levels of aliphatic (**A**) and indole (**B**) glucosinolate components and expression of biosynthesis genes under four specific treatments (C, control; M1, mock day 1; T1, treated day 1; M3, mock day 3; T3, treated day 3) in white-mold-inoculated R (SCNU-C-049) and S (SCNU-C-033) lines. Blue and bold letters represent statistically significant correlations (*p* < 0.05). For each gene and glucosinolate combination, the values indicate the Pearson correlation coefficient. Red cells represent positive correlation and green cells represent negative correlation. Yellow cells represent no significant correlation. Glucosinolate (GSL) components: GIB, glucoiberin; PRO, progoitrin; GRA, glucoraphanin; SIN, sinigrin; GNA, gluconapin; GIV, glucoiberverin; GER, glucoerucin; GBS, glucobrassicin; NGBS, neoglucobrassicin; MGBS, methoxyglucobrassicin; HGBS, hydroxyglucobrassicin. R, resistant; S, susceptible.

**Figure 6 ijms-19-04037-f006:**
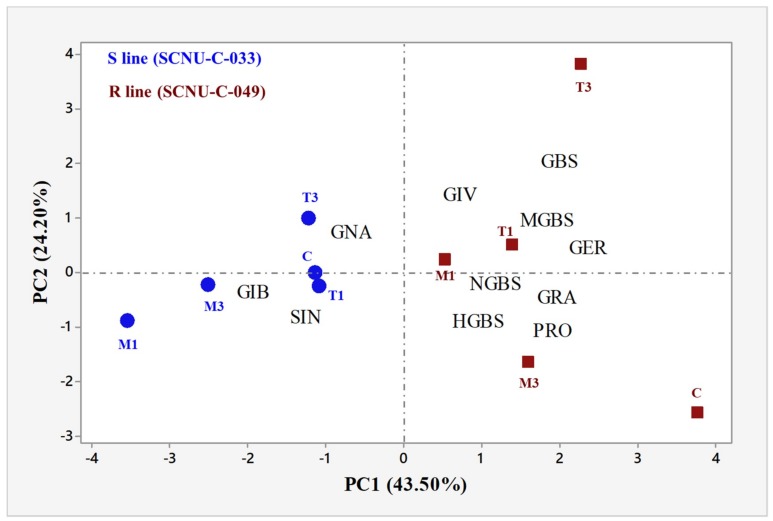
Biplot of white-mold-inoculated R (SCNU-C-049) and S (SCNU-C-033) cabbage lines and individual glucosinolate components, as determined by principal component analysis (PCA). Dark red squares denote mean PC scores of the R line, and blue circles those of the S line. Fungal treatments: C, control; M1, mock day 1; T1, treated day 1; M3, mock day 3; T3, treated day 3. Glucosinolate (GSL) components: GIB, glucoiberin; PRO, progoitrin; GRA, glucoraphanin; SIN, sinigrin; GNA, gluconapin, GIV, glucoiberverin; GER, glucoerucin; GBS, glucobrassicin; NGBS, neoglucobrassicin; MGBS, methoxyglucobrassicin; HGBS, hydroxyglucobrassicin. R, resistant; S, susceptible.

**Figure 7 ijms-19-04037-f007:**
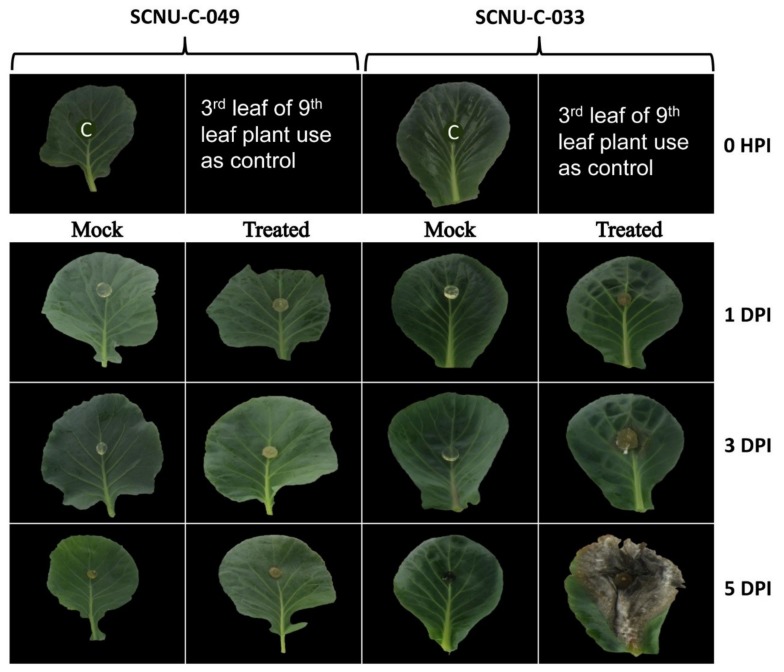
White mold disease progress in resistant (R; SCNU-C-049) and sensitive (S; SCNU-C-033) lines of cabbage. The third-youngest leaf was infected at the of ninth leaf stage of each plant. Infected leaves were examined at 0 hours post inoculation (HPI), and then from 1 to 5 days post inoculation (DPI). C, control plant leaf (no inoculation). ‘Mock’ treatment(s) - were those done with mycelium-free agar plugs.

**Table 1 ijms-19-04037-t001:** Differential expression of genes related to the glucosinolate biosynthesis pathway in white-mold-inoculated R and S cabbage. Numbers in blue and purple color letter indicate days at which genes were expressed and fold differences compare to mock-treated plants, respectively. R, resistant (SCNU-C-049); S, susceptible (SCNU-C-033).

Function of Gene Product	Genes Upregulated in R Line	Genes Upregulated in S Line	Genes Upregulated in both R and S Lines	
			Higher in R Line	Higher in S Line
**Transcription Factor**	*MYB28-Bol017019* (1, 10.2); *MYB34-Bol017062* (1, 3.08)	*MYB28-Bol036743* (1, 5.8); *MYB29-Bol008849* (1, 51.0); *MYB28-Bol036286* (1, 21.6); *MYB28-Bol007795* (1, 2.9)	*MYB122-Bol026204* (3, 189.1)	
**Aliphatic Biosynthesis**	*ST5b-Bol026202* (3, 5.6); *ST5c-Bol030757* (3, 68.6)	*AOP2-Bo3g052110* (1, 8.4); *AOP2-Bo9g006240* (1, 11.4); *FMOGS-OX2-Bol010993* (1, 5.5); *GSL-OH-Bol033373* (1, 65.3)		
Indole Biosynthesis	*ST5a-Bol039395* (3, 25.3); *ST5a-Bol026200* (3, 82.9)	*CYP81F1-Bol017375* (1, 2.0); *CYP81F1-Bol017376* (3, 7.96)	*CYP81F4-Bol032712* (3, 21.1); *CYP81F2-Bol026044* (3, 54.5); *CYP81F2-Bol014239* (3, 871); *CYP81F2-Bol012237* (3, 48.3); *IGMT1-Bol007029* (3, 24.7); *IGMT2-Bol007030* (3, 171)	*CYP81F1-Bol028914* (3, 19.6); *CYP81F4-Bol032714* (3, 12.3); *IGMT1-Bol020663* (3, 652)

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
