# Peer review of "Glucosinolate Profiling and Expression Analysis of Glucosinolate Biosynthesis Genes Differentiate White Mold Resistant and Susceptible Cabbage Lines"

_ijms, 2018, doi:10.3390/ijms19124037_

Round 1
Reviewer 1 Report
The manuscript identified a mold resistant line, and compared glucosinolate profiling and related gene in the resistant line and susceptible line undertreated condition. However, some corrections or detail information was needed before publishing.
In row 109-110, authors say "The same pattern of results was obtained in the case of stem infection for the same 45 inbred lines (data not shown)". If authors feel the results were not related with the manuscript, just remove this sentence, or show the data.
In table 1 or row 180, authors say "Numbers in green and red indicate days...", however, I can not find the colored number or word in the table. Please check and correct
In figure 5, the color is so confused, authors say "Red cells represent positive correlation and green cells represent negative correlation", but how about yellow.
In row 270-272, the discussion should have more information, not just to compare present study and previous study, authors should give more information about the comparison, such as what author want to show by the comparison.
In row 309, Please correct the wording? Control? The original paper just hypothesis that and aliphatic glucosinolate biosynthesis controls the response to biotic challenges or the MYB28.
In row 380-383: How many plants in one large pot? is the environment keep same after transplant?
Are they reached the ninth-leaf stage at a similar date? I saw that the RH was changed after inoculated, is the RH affect the disease happen? please show the reason.
In row 392, author say "three replicates from each ...", please clarify: were the three replicates on the same plant or on three different plants.
In row 394, authors say "with modification", please clarify how to modification
In row 398-400, Not clear, one line was measured in 1 to 5 days with five measurements, or all lines were measured one time from 1-5 days? And inconsistent with above “they were evaluated for white mold disease on month after head formation”
In row 398-400, are the two procedures use the same plants?
In row 417, when was the sample collected? how long time after infected? which leaf was collected? Whole leaf or piece of the leaf was collected? Authors should give more detail information.
Author Response
Response to Reviewer 1 comments
Comments: In row 109-110, authors say "The same pattern of results was obtained in the case of stem infection for the same 45 inbred lines (data not shown)". If authors feel the results were not related with the manuscript, just remove this sentence, or show the data.
Responses: Thanks for good suggestions, we have removed this sentence from that sections.
Comments: In table 1 or row 180, authors say "Numbers in green and red indicate days...", however, I cannot find the colored number or word in the table. Please check and correct
Responses: We have now replaced ‘green and red’ with ‘blue and purple’, respectively to make those visible to all. Thanks
Comments: In figure 5, the color is so confused, authors say "Red cells represent positive correlation and green cells represent negative correlation", but how about yellow.
Responses: We try to resolve the confusion by replacing red letters to blue-bold letters in Figure 5 and as well as in supplementary file 3. Yellow cells re-present no significant correlation.
Comments: In row 270-272, the discussion should have more information, not just to compare present study and previous study, authors should give more information about the comparison, such as what author want to show by the comparison.
Responses: Paragraph revised. Following sentences inserted:
The genotypes screened in this study were developed through breeding. Thus the observed differences explain genotypic variation and indicate that resistance towards S. sclerotiorum is genotype specific. Therefore, the factors controlling the resistance could be transferred to the elite cabbage lines.
Comments: In row 309, Please correct the wording? Control? The original paper just hypothesis that and aliphatic glucosinolate biosynthesis controls the response to biotic challenges or the MYB28.
Responses: Sentence revised
In Arabidopsis, biotic challenges are believed to be regulated by the upregulation of MYB28, a novel regulator of aliphatic glucosinolate biosynthesis genes [60].
Comments: In row 380-383: How many plants in one large pot? is the environment keep same after transplant?
Responses: One plant per pot was kept after transplantation. Yes, plants were kept at the same culture condition until inoculation was imposed.
Comments: Are they reached the ninth-leaf stage at a similar date? I saw that the RH was changed after inoculated, is the RH affect the disease happen? Please show the reason.
Responses: We had 15 seedlings against each genotype. We used hat plants for inoculation which reached ninth leaf at the day of inoculation. High RH positively influence white mold disease development (see Harikrishnan et al. 2006).
Harikrishnan, R., and L. E. Del Río. "Influence of temperature, relative humidity, ascospore concentration, and length of drying of colonized dry bean flowers on white mold development." Plant Disease 90.7 (2006): 946-950
Comments: In row 392, author say "three replicates from each ...", please clarify: were the three replicates on the same plant or on three different plants.
Responses: We used three different plants for three replications.
Comments: In row 394, authors say "with modification", please clarify how to modification
Responses: Comparison with Zhao and Meng 2003 and Yu et al. 2010 (see supplementary table S9).
Zhao and Meng 2003
| Yu et al. 2010 | Our study |
Detached leaf was infected at 9 to 12 leaf stages. | Detached leaves were infected. | Inoculated on live leaves at 9th leaf stage |
Two pieces of mycelial agar plug (5 mm in diameter) were used for infection | A mycelial agar plug (8 mm in diameter) was used for infection | One piece of mycelial plug (7 mm in diameter) was used for infection |
Stems were inoculated one month before the sampling using a tooth-pick | Stems were inoculated two weeks after final flowering | Stems were inoculated at 9th leaf with alpin |
Inoculation was done at a height of 35 cm above the ground | Inoculation was done at a height of 50 cm above the ground | Inoculation was done at a height of 20 cm above the ground |
Infected lesion was measured at 5 days post inoculation (DPI) | Infected lesion was measured at 7 DPI | The lesion was measured from 0 to 5 DPI |
The stem tooth-pick method was used for assessing the resistance in adult plants. | Cabbage heads were inoculated one month after head formation and then observed the head lesions at 10 DPI (with the head right side up) and at 15 DPI (with the head upside down). |
Comments: In row 398-400, Not clear, one line was measured in 1 to 5 days with five measurements, or all lines were measured one time from 1-5 days? And inconsistent with above “they were evaluated for white mold disease on month after head formation”
Responses: All lines were measured three times from 1-5 days after inoculation. We inoculated all lines one month after head formation and then observed head lesions at 10 and 15 DAIs.
Comments: In row 398-400, are the two procedures use the same plants?
Responses: Different plants are used for two procedures
Comments: In row 417, when was the sample collected? how long time after infected? which leaf was collected? Whole leaf or piece of the leaf was collected? Authors should give more detail information.
Responses: Leaf samples at 1 and 3 DPI were taken. Inoculated but non-infected and mock-treated leaf pieces and were collected at sampling.
Regards and thanks,
Professor Ill-Sup Nou
Corresponding Author
Department of Horticulture
Sunchon National University, South Korea
Email: nis@sunchon.ac.kr
Reviewer 2 Report
ijms-360364
Full Title: Glucosinolate profiling and expression analysis of biosynthesis genes differentiate white mold resistant and susceptible cabbage lines
General comments to the manuscript:
The present manuscript describes a study that comes in the line of a previous work described by the group aiming to understand the molecular mechanism behind cabbage plant tolerance upon Sclerotina sclerotiorum infection. The authors here describe the expression pattern of genes involved in the glucosinolates secondary metabolites biosynthesis, and complement those results with metabolomics data by quantifying specific metabolites in association upon fungal infection.
The manuscript is very well written, it comprises a good introduction, the results are very interesting and would merit of publication in the International Journal of Molecular Sciences journal; however, there are some few points that need to be improved before final acceptance, which I think that will contribute to increase the quality of manuscript. I would recommend to improve the manuscript making the changes suggested below.
I agree with major revision.
Title: I would suggest to change the title in order to be clearer that gene expression analysis was made in the genes involved in the Glucosinolates biosynthesis.
Results
In section 2.3 please introduce the work done, as example: “In order to investigate whether the expression level of genes belonging to the aliphatic and indole GSL biosynthesis and their associated transcription factors are associated with plant resistance upon S. sclerotionum infection, reverse transcription quantitative PCR (RT-qPCR) was performed. “ After this sentence introduce the information regarding the genes used as reported genes, like “Three Actin genes (include the names) were used for normalization of expression data and 38 target genes were considered for analysis.”
Line 166: Table 2 is missing.
Line 167: Rewrite last sentence in order to be clearer that were only the Actin genes the genes used as reference genes.
Regarding the expression analysis, there are some points that I would like to ask for clarification:
- considering the limitations of the technique and that the most appropriate amplicon size must be smaller than 100bp what was the reason to design primers considering so large amplicons, in some cases of 500bp?
- for expression data normalization the authors report the use of three Actin genes and the application of the Livak’s comparative 2-∆∆Ct method (in materials and methods section, line 465). Can the authors explain the procedure of using three genes with the method described? Did the authors tried to select the most appropriate pair of genes using the GeNorm software to then proceed with data normalization?
- introduce in Table S1 the accession number of each gene (it is not clear if this information is already considered in the Table; as example, the reference B01007795 is the accession number?
- Include in the Table S1 the information regarding primers efficiency and R2. It must be included in the material and methods section the procedure followed to assess primers efficiency. Have the authors considered the standard curve to assess primers efficiency?
- Regarding primers specificity the authors must explain how they confirm that primers amplified the target gene. Regarding melting curve analysis (Supplementary file 4) it will be required the introduction of the negative controls and the information of the samples considered on the graphs.
- An high important point that can be seen in the graphs of some genes is the existence of differences in the melting curves, which would mean the amplification of different amplicons. This can be seen in the graph of Bol017019, Bol008849, Bol026204, Bol039395, Bol031350, Bol026044. It would be important to provide a agarose gel showing the amplicon generated by each primers pair.
In Fig. 2 please include the name of the genes as the acronym used in the text. In that way it will be much easier to follow the data description.
In Table 1, Line 180 “Numbers in green and red”, this information is missing in the table.
Line 216 “…we observed inconsistent responses in the expression of 12 genes…” In these 12 genes there are included some that primers seems to present no specificity (seen by differences in melting curves), and that could be the reason for that results. It will be extremely important to clarify the questions related with primers specificity and efficiency to then take right conclusion from results.
Line 248: Rewrite sentence, as example “ These results show the association between expression of genes involved in GSLs biosynthesis and changes in the GSL metabolite profile.”
Regarding PCA analysis it is not clear the statistical analysis performed. In the legend it appear the reference to R and S lines but none appear in the graph; additionally, the name of the metabolites appear in three different colours (black, red and pink), what the meaning of that difference? In materials and methods subsection 4.8. Statistical analysis, it must be included the aim and the procedure followed to perform the PCA analysis.
Discussion
Line 324-325: it will be interesting if authors better discuss the idea related with the no link between gene expression and phenotype, or if the genes upregulated in the S line are not key genes in the biosynthetic pathway.
Line 333: In my point of view the authors could better explore the idea related with the signalling pathway associated with S. sclerotiorum. It is quite known that infection is perceived by plant by a signalling pathway. The nature of the signalling pathway could depends on the fungus nature, if biotrophic or necrotrophic. Considering the nature of the S. sclerotiorum and the results achieved by gene expression it will be interesting to discuss the data giving a focus on the signalling pathway.
Line 351: develop a bit better the sentence, at least it will be interesting to know if the data achieved by the authors of (55) were achieved in the same plant species and/or pathogenic fungus.
Materials and methods
Present in a separated sub-section the experimental design followed for metabolite and gene expression analysis; please provide the difference between control and mock-treated plants, and explain the meaning of replicates and its composition (does one replicate means a single plant?).
The subsection 4.6 must consider not only primers design but also the methodology followed to assess primers specificity and efficiency.
In the subsection 4.7 please clarify if integrity and purity of total RNA was not evaluated. Include the concentration of total RNA used for cDNA syhthesis; change the volume of the PCR components by its concentration. Include the information regarding biological and technical replicates done, and the controls included.
Please explain the sentence on line 454 “and the quality of cDNA was standardized with primers derived from three B. oleracea actin genes of (Supplementary file 5).”
Regarding the subsection 4.8. Statistical analysis I would like to ask for the ANOVA table in which must be included the results from that statistical analysis, including each level of variability (R line, S line, control, mock-treated, inoculated, and the timepoints) and the interaction between the different levels. The Table S6 and S7 are not giving the results of the ANOVA but only the p-values. It is not possible to understand the levels that were considered for the analysis, were the timepoints not considered? The interactions must appear with X and not with &.
Author Response
Response to Reviewer 2 comments
General comments to the manuscript:
Comments: The present manuscript describes a study that comes in the line of a previous work described by the group aiming to understand the molecular mechanism behind cabbage plant tolerance upon Sclerotina sclerotiorum infection. The authors here describe the expression pattern of genes involved in the glucosinolates secondary metabolites biosynthesis, and complement those results with metabolomics data by quantifying specific metabolites in association upon fungal infection. The manuscript is very well written, it comprises a good introduction, the results are very interesting and would merit of publication in the International Journal of Molecular Sciences journal; however, there are some few points that need to be improved before final acceptance, which I think that will contribute to increase the quality of manuscript. I would recommend to improve the manuscript making the changes suggested below.
Responses: Thanks for your compliments. We have tried to accommodate your valuable suggestions.
Comments: Title: I would suggest to change the title in order to be clearer that gene expression analysis was made in the genes involved in the Glucosinolates biosynthesis.
Responses: Revised
Glucosinolate profiling and expression analysis of glucosinolate biosynthesis genes differentiate white mold resistant and susceptible Cabbage lines
Comments: In section 2.3 please introduce the work done, as example: “In order to investigate whether the expression level of genes belonging to the aliphatic and indole GSL biosynthesis and their associated transcription factors are associated with plant resistance upon S. sclerotionum infection, reverse transcription quantitative PCR (RT-qPCR) was performed. “ After this sentence introduce the information regarding the genes used as reported genes, like “Three Actin genes (include the names) were used for normalization of expression data and 38 target genes were considered for analysis.”
Responses: Following sentences inserted as suggested at the beginning of the section 2.3.
In order to investigate whether the expression level of genes belonging to the aliphatic and indole GSL biosynthesis and their associated transcription factors are associated with plant resistance upon S. sclerotionum infection, reverse transcription quantitative PCR (RT-qPCR) was performed. Three Actin genes (actin1, actin2, and actin3, see section 4.7) were used for normalization of expression data and 38 target genes were considered for analysis.
Comments: Line 166: Table 2 is missing.
Responses: It is would be Table 1. We have corrected it now. Thanks
Comments: Line 167: Rewrite last sentence in order to be clearer that were only the Actin genes the genes used as reference genes.
Responses: corrected accordingly. Thanks
Comments: - Regarding the expression analysis, there are some points that I would like to ask for clarification:
considering the limitations of the technique and that the most appropriate amplicon size must be smaller than 100bp what was the reason to design primers considering so large amplicons, in some cases of 500bp?
Responses: We have published a series of papers with this primer sets. We had to design some primers with large amplicon size to obtain acceptable melting curves.
For example,
Robin, A.H.K.; Yi, G.-E.; Laila, R.; Yang, K.; Park, J.-I.; Kim, H.R.; Nou, I.-S. Expression Profiling of Glucosinolate Biosynthetic Genes in Brassica oleracea L. var. capitata Inbred Lines Reveals Their Association with Glucosinolate Content. Molecules 2016, 21, 787. doi: 10.3390/molecules21060787
Abuyusuf, M.; Robin, A.H.K.; Kim, H.-T.; Islam, M.R.; Park, J.-I.; Nou, I.-S. Altered Glucosinolate Profiles and Expression of Glucosinolate Biosynthesis Genes in Ringspot-Resistant and Susceptible Cabbage Lines. Int. J. Mol. Sci. 2018, 19, 2833.
Yi, G.-E.; Robin, A.H.K.; Yang, K.; Park, J.-I.; Hwang, B.H.; Nou, I.-S. Exogenous methyl jasmonate and salicylic acid induce subspecies-specific patterns of glucosinolate accumulation and gene expression in Brassica oleracea L. Molecules 2016, 21, 1417.
Robin, A.H.K.; Yi, G.-E.; Laila, R.; Hossain, M.R.; Park, J.-I.; Kim, H.R.; Nou, I.-S. Leptosphaeria maculans alters glucosinolate profiles in blackleg disease-resistant and-susceptible cabbage lines. Frontiers in Plant Science 2017, 8, 1769.
Yi, G.E., Robin, A.H.K., Yang, K., Park, J.I., Kang, J.G., Yang, T.J. and Nou, I.S., 2015. Identification and expression analysis of glucosinolate biosynthetic genes and estimation of glucosinolate contents in edible organs of Brassica oleracea subspecies. Molecules, 20(7), pp.13089-13111.
Comments: - for expression data normalization the authors report the use of three Actin genes and the application of the Livak’s comparative 2-∆∆Ct method (in materials and methods section, line 465). Can the authors explain the procedure of using three genes with the method described? Did the authors tried to select the most appropriate pair of genes using the GeNorm software to then proceed with data normalization?
Responses: We obtained an average Cq for three actin genes before data normalization. We did not use GeNorm software. We have followed similar procedure in our previous publications.
Comments: - introduce in Table S1 the accession number of each gene (it is not clear if this information is already considered in the Table; as example, the reference B01007795 is the accession number?
Responses: Yes, Bol007795 is an accession number for MYB28.
Now genes and accessions are separated in Table S1.
Comments: - Include in the Table S1 the information regarding primers efficiency and R2. It must be included in the material and methods section the procedure followed to assess primers efficiency. Have the authors considered the standard curve to assess primers efficiency?
Responses: Yes, standard curves were assessed to calculated primer efficiencies following Robin et al. (2016),
To test the efficiency of the primers, the pooled cDNA of equal concentrations, 300 ng·µL−1, of cabbage inbred lines was serially diluted by 10-fold, and 1.0 µL of the 100, 10−1, 10−2, 10−3, 10−4 and 10−5 diluted cDNA was used as the template in each reaction with forward and reverse primers. The qRT-PCR reaction was performed for 40 cycles along with the melting curve in triplicate along with no template control. A semi-log plot for Ct versus fold dilution was drawn to find out the slope. The efficiency of the primer was calculated by using the following formula, e = 10^(−1/slope). A primer with −3.321928 slopes was found 100% efficient. Primers with an 85%–100% efficiency level were chosen for expression analysis, otherwise redesigned to obtain the efficiency within the expected range.
Comments: - Regarding primers specificity the authors must explain how they confirm that primers amplified the target gene. Regarding melting curve analysis (Supplementary file 4) it will be required the introduction of the negative controls and the information of the samples considered on the graphs.
Responses: We have carried out qPCR with and without DNA templates (1 micro L ultra pure water was used as a negative control). There was no peak for the negative controls. For example, see attached file 1.
Comments: - An high important point that can be seen in the graphs of some genes is the existence of differences in the melting curves, which would mean the amplification of different amplicons. This can be seen in the graph of Bol017019, Bol008849, Bol026204, Bol039395, Bol031350, Bol026044. It would be important to provide a agarose gel showing the amplicon generated by each primers pair.
Responses: Agarose gel electrophoresis of Bol017019, Bol008849, Bol026204, Bol039395, Bol031350, and Bol026044 genes were conducted. No additional amplicons were observed. We believe that provides reliability of our results. (see attached figure 2)
Comments: In Fig. 2 please include the name of the genes as the acronym used in the text. In that way it will be much easier to follow the data description.
Responses: Thanks for your advice.
Comments: In Table 1, Line 180 “Numbers in green and red”, this information is missing in the table.
Responses: We realize your point. Now we move this sentence to the title of table 1
Comments: Line 216 “…we observed inconsistent responses in the expression of 12 genes…” In these 12 genes there are included some that primers seems to present no specificity (seen by differences in melting curves), and that could be the reason for that results. It will be extremely important to clarify the questions related with primers specificity and efficiency to then take right conclusion from results.
Responses: Primer specificity rechecked. Thanks
Comments: Line 248: Rewrite sentence, as example “ These results show the association between expression of genes involved in GSLs biosynthesis and changes in the GSL metabolite profile.” Regarding PCA analysis it is not clear the statistical analysis performed. In the legend it appear the reference to R and S lines but none appear in the graph; additionally, the name of the metabolites appear in three different colours (black, red and pink), what the meaning of that difference?
Responses: Sentence deleted.
Black means inconsistent response by individual glucosinolates, red indicates increasing trends in R line and pink means increasing trends in S line. To remove this ambiguity we just change the red and pink into black color.
Comments: In materials and methods subsection 4.8. Statistical analysis, it must be included the aim and the procedure followed to perform the PCA analysis.
Responses: Sentence inserted:
A principal component analysis with correlation matrix command was conducted using Minitab 18 (State College, PA, USA) statistical software to explore association between glucosinolate contents after S. sclerotiorum inoculation and resistance of cabbage lines.
Discussion
Comments: Line 324-325: it will be interesting if authors better discuss the idea related with the no link between gene expression and phenotype, or if the genes upregulated in the S line are not key genes in the biosynthetic pathway.
Responses: Sentence revised:
In contrast, a number of genes were highly expressed only in the S line after inoculation indicating that these genes are either not the key regulators or any altered glucosinolate by them (for example, gluconapin) has no important role in defense against S. sclerotiorum infection
Comments: Line 333: In my point of view the authors could better explore the idea related with the signalling pathway associated with S. sclerotiorum. It is quite known that infection is perceived by plant by a signalling pathway. The nature of the signalling pathway could depends on the fungus nature, if biotrophic or necrotrophic. Considering the nature of the S. sclerotiorum and the results achieved by gene expression it will be interesting to discuss the data giving a focus on the signalling pathway.
Responses: Sentence revised.
Since the nature of the signalling pathway could depends on the fungus type and the S. sclerotiorum is a necrotrophic fungus therefore it is likely that jasmonic acid (JA), ethylene (ET), and salicylic acid (SA), signalling pathway might be involved in resistance response of cabbage lines (Alkooranee et al 2015, and 2015a).
References inserted:
Alkooranee JT, Aledan TR., Xiang J, Lu G and Li M. Induced Systemic Resistance in Two Genotypes of B. napus and R. alboglabra (RRCC) by Trichoderma Isolates against S. sclerotiorum. American Journal of Plant Sciences. 2015, 6, 1662–1674.
Alkooranee JT, Yin Y, Aledan TR, Jiang Y, Lu G, Wu J, et al. Systemic Resistance to Powdery Mildew in Brassica napus (AACC) and Raphanus alboglabra (RRCC) by Trichoderma harzianum TH12. PLoS ONE, 2015. 10 (11).
Comments: Line 351: develop a bit better the sentence, at least it will be interesting to know if the data achieved by the authors of (55) were achieved in the same plant species and/or pathogenic fungus.
Responses:
What plant species? Arabidopsis
What fungus? Blumeria graminis Erysiphe pisi and Plectosphaerella cucumerina
What was found? CYP81F2 with 4MI3G biosynthesis showed impaired entry resistance to Blumeria graminis and Erysiphe pisi and were more susceptible to Plectosphaerella cucumerina
Revised sentence: Our findings also agree with the report that in Arabidopsis upon infection with Blumeria graminis Erysiphe pisi and Plectosphaerella cucumerina CYP81F2 expression induces antifungal defenses [55].
Materials and methods
Comments: Present in a separated sub-section the experimental design followed for metabolite and gene expression analysis;
Response: Samples were harvested from same experiment for metabolite and gene expression analysis.
Comments: please provide the difference between control and mock-treated plants,
Responses: Control plants remained undisturbed whereas mock-treatment was done with mycelium-free agar plugs.
Comments: explain the meaning of replicates and its composition (does one replicate means a single plant?).
Responses: Biological replicates. Each sample was harvested from a single plant.
Comments: The subsection 4.6 must consider not only primers design but also the methodology followed to assess primers specificity and efficiency.
Responses: Please see previous response.
Comments: In the subsection 4.7 please clarify if integrity and purity of total RNA was not evaluated. Include the concentration of total RNA used for cDNA syhthesis; change the volume of the PCR components by its concentration. Include the information regarding biological and technical replicates done, and the controls included. Please explain the sentence on line 454 “and the quality of cDNA was standardized with primers derived from three B. oleracea actin genes of (Supplementary file 5).”
Responses: Sentence rewritten
Complementary DNA (cDNA) was synthesized by using 5 µg of extracted RNA and oligo (dT) primer of first strand cDNA synthesis kit (Invitrogen, Madison, WI, USA) following the manufacturer’s instructions and the equality of cDNA for the samples was normalized by comparing the thickness of the PCR amplicons from one B. oleracea actin genes.
There were three technical replicates for each biological replicate.
Comments: Regarding the subsection 4.8. Statistical analysis I would like to ask for the ANOVA table in which must be included the results from that statistical analysis, including each level of variability (R line, S line, control, mock-treated, inoculated, and the time points) and the interaction between the different levels. The Table S6 and S7 are not giving the results of the ANOVA but only the p-values. It is not possible to understand the levels that were considered for the analysis, were the time points not considered? The interactions must appear with X and not with &.
Responses: Responses: What we are actually looking at variation among treatment-timepoint combinations (5 combinations) within resistant and susceptible lines (2 genotypes). In graphs we have 10 bars of such kind. An oneway ANOVA with such data have 9 dfs what are presented in Table S6 and Table S7.
1 | R-C | |
2 | R-M1 | |
3 | R-T1 | |
4 | R-M3 | |
5 | R-T3 | |
6 | S-C | |
7 | S-M1 | |
8 | S-T1 | |
9 | S-M3 | |
10 | S-T3 |
C, control; M1, mock day 1; T1, treated day 1; M3, mock day 3; T3, treated day 3.
R, resistant (SCNU-C-049); S, susceptible (SCNU-C-033).
A separate ANOVA has been conducted with Genotype (Resistant versus susceptible), treatment-timepoint (5 combinations) and Genotype x treatment-timepoint interactions to fulfil your curiosity (see an example below),
Thanks for your scholastic review.
Regards and thanks,
Professor Ill-Sup Nou
Corresponding Author
Department of Horticulture
Sunchon National University, South Korea
Email: nis@sunchon.ac.kr
General Linear Model: MYB28-Bol007, MYB28-Bol036, ... versus ResVsus, Treat-Tim
Factor Type Levels Values
ResVsus fixed 2 1, 2
Treat-Time fixed 5 1, 2, 3, 4, 5
Analysis of Variance for MYB28-Bol007795, using Adjusted SS for Tests
Source DF Seq SS Adj SS Adj MS F P
ResVsus 1 17.8726 17.8726 17.8726 836.60 0.000
Treat-Time 4 18.6040 18.6040 4.6510 217.71 0.000
ResVsus*Treat-Time 4 27.8713 27.8713 6.9678 326.16 0.000
Error 20 0.4273 0.4273 0.0214
Total 29 64.7751
S = 0.146162 R-Sq = 99.34% R-Sq(adj) = 99.04%
Least Squares Means
MYB28-Bol007795 MYB28-Bol036286 MYB28-Bol017019 MYB28
ResVsus Mean SE Mean Mean SE Mean Mean SE Mean Mean
1 0.736 0.03774 1.480 0.15585 1.204 0.02457 0.538
2 2.280 0.03774 9.529 0.15585 0.019 0.02457 2.222
Treat-Time
1 1.331 0.05967 5.137 0.24641 0.501 0.03885 1.542
2 1.445 0.05967 2.679 0.24641 0.231 0.03885 1.118
3 2.892 0.05967 15.844 0.24641 2.273 0.03885 2.901
4 1.433 0.05967 2.898 0.24641 0.027 0.03885 1.268
5 0.439 0.05967 0.965 0.24641 0.024 0.03885 0.070
ResVsus*Treat-Time
1 1 1.000 0.08439 1.000 0.34848 1.000 0.05494 1.000
1 2 0.965 0.08439 3.914 0.34848 0.441 0.05494 1.237
1 3 0.214 0.08439 0.449 0.34848 4.502 0.05494 0.039
1 4 1.395 0.08439 1.185 0.34848 0.050 0.05494 0.333
1 5 0.107 0.08439 0.854 0.34848 0.026 0.05494 0.081
2 1 1.662 0.08439 9.274 0.34848 0.003 0.05494 2.084
2 2 1.925 0.08439 1.445 0.34848 0.021 0.05494 0.999
2 3 5.570 0.08439 31.238 0.34848 0.043 0.05494 5.764
2 4 1.472 0.08439 4.611 0.34848 0.004 0.05494 2.203
2 5 0.770 0.08439 1.077 0.34848 0.022 0.05494 0.059

Round 2
Reviewer 1 Report
After revision, authors addressed all my questions or comments.
It will be great if authors can integrate the response in the manucript will great: "We had 15 seedlings against each genotype. We used hat plants for inoculation which reached ninth leaf at the day of inoculation."
Author Response
Response to Reviewer 1- Round 2
After revision, authors addressed all my questions or comments.
Comments: It will be great if authors can integrate the response in the manuscript: "We had 15 seedlings against each genotype. We used that plants for inoculation which reached ninth leaf at the day of inoculation."
Responses: Thanks for nice suggestions, we have inserted following sentence in the manuscript, “We reared 15 seedlings against each genotype to obtain seedlings of homogenized growth at the day of inoculation. We used plants those reached ninth leaf at the day of inoculation.”
Thanks for your scholastic review. Your comments helped us improve this manuscript.
Regards and thanks,
Professor Ill-Sup Nou
Corresponding Author
Department of Horticulture
Sunchon National University, South Korea
Email: nis@sunchon.ac.kr
Reviewer 2 Report
ijms-385957-peer-review-v2
Full Title: Glucosinolate profiling and expression analysis of Glucosinolate biosynthesis genes differentiate white mold resistant and susceptible cabbage lines
General comments to the manuscript:
The results described in this manuscript are interesting and would merit of publication in the International Journal of Molecular Sciences journal. It is easily seen that the manuscript was improved by authors taking in consideration the reviewer’s comments, however there are several points that still need to be revised and improved. I would recommend to improve the manuscript making the changes suggested below. A revision of the language must be considered due to the existence of some typographical errors.
I still agree with major revision.
Main aspects to be considered for revision:
Line 160: “The expression levels and melting curves of three actin genes, reference genes, and 38 glucosinolate-biosynthesis-related genes were used for standardization and are given in supplementary file 3 and supplementary file 4.” This sentence must be rewritten in order to clarify what were the reference genes (actin genes) and the genes used for normalization. Besides that, I couldn’t find the expression data of the Actin genes in supplementary files.
Regarding the reference genes I would like to ask authors to select the most appropriate pair of primers, from the three actin genes, to perform normalization using the geNorm software. I have doubts about the stability of Actin 3.
Primers efficiency was included in Table S1 but R2 is still missing. The information included allow us to see efficiencies below 90%, the worst cases are Bol028919 and Bol032711 with the lowest efficiency (85%). In my point of view it would be required the design new primer pairs for all cases that showed efficiency below 90%.
The information regarding the Bol028918 gene is missing.
In my point of view there are results of gene expression analysis that remain not acceptable. The variability that I previously reported was not checked properly. The authors evaluate primers specificity by running the PCR product (it is not clear if from real-time or end-point PCR) in an agarose gel, and the observation of a single band was sufficient to them to conclude that primers were specific. However, primers could amplify different fragment with very similar sizes but with differences in the sequence, which could be related with the variability observed. The worst cases are the genes FMOGS-OX5-Bol031350 and CYP81F2-Bol026044, which in my point of view are not acceptable.
In page 4: The PCA analysis remains for me very strange. I cannot understand why there are timepoints and glucosinolate components mixed, and also the reason why there are more samples in red than in blue.
Line 343: “Since the nature of the signaling pathway could depends on the fungus type and the S. sclerotiorum is a necrotrophic fungus therefore it is likely that jasmonic acid (JA), ethylene (ET), and salicylic acid (SA), signaling pathway might be involved in resistance response of cabbage lines [62,63].” In my point of view this topic could be much better discussed. The authors are presenting the three signaling pathways, but it is known that one of those is more involved in plant response upon necrotrophic fungus. It will be interesting to have that information here presented and discussed.
Line 398: “inoculation chamber” change it to growth chamber.
Regarding the sampling for gene expression analysis and metabolite analysis, it must be clear in the text that the sampling was made simultaneously.
Line 409: Rephrase the sentence “Three biological replicates from three different plants of each of 45 inbred lines were inoculated into the third-youngest leaf of ninth-leaf-stage plants.” to “Three different plants of each of the 45 inbred lines (biological replicates) were inoculated into the third-youngest leaf of ninth-leaf-stage plants”.
Line 436: “from three randomly selected plants of the R line and the S line” The procedure followed remains unclear. If only three plants were inoculated how was the selection made randomly?
Line 475: “from one B. oleracea actin genes (Supplementary file 5).” The information regarding the gene evaluated is not given and also the PCR conditions not presented in materials and methods section. Additionally it seems to be present a larger fragment around 500pb (?), which could be related with no primers specificity, also seen in melting curve of Actin3.
Line 479: primers concentration was not included and remains the volume added to PCR mix.
The information regarding the evaluation of RNA quality is still not included.
The suggested changes for section 4.6 were not all considered. The information regarding primers specificity was not here included.
In Line 465: “their efficiencies were tested” rewrite the sentence. Consider that efficiencies were not tested but calculated using parameters achieved by standard curve.
At Supplementary file 5: RNA purity (260/280) is much lower than 2.0 which could lead to variability at the final results, not only do to RNA degradation (was this evaluated?) but also do to the presence of inhibitors at the solution.
Author Response
Response to Reviewer 2- Round 2
ijms-385957-peer-review-v2
Full Title: Glucosinolate profiling and expression analysis of Glucosinolate biosynthesis genes differentiate white mold resistant and susceptible cabbage lines
General comments to the manuscript:
Comments: The results described in this manuscript are interesting and would merit of publication in the International Journal of Molecular Sciences journal. It is easily seen that the manuscript was improved by authors taking in consideration the reviewer’s comments, however there are several points that still need to be revised and improved. I would recommend to improve the manuscript making the changes suggested below. A revision of the language must be considered due to the existence of some typographical errors.
Responses: Thanks for your compliments towards our efforts. We have revised the manuscript again according to your suggestions.
Comment-1: Line 160: “The expression levels and melting curves of three actin genes, reference genes, and 38 glucosinolate-biosynthesis-related genes were used for standardization and are given in supplementary file 3 and supplementary file 4.” This sentence must be rewritten in order to clarify what were the reference genes (actin genes) and the genes used for normalization.
Besides that, I couldn’t find the expression data of the Actin genes.
Responses: Sentences revised:
The Actin gene (actin1) was used for normalization of expression of 38 target genes. The expression levels and melting curves of Actin gene and 38 glucosinolate-biosynthesis-related genes are given in Supplementary file 3 and 4, respectively. The primer of actin1 was designed from the sequence obtained from the NCBI database (GenBankAccession no. AF044573) [35, 58] since that gene was expressed in both the R and S lines and was used as a reference.
Expression data of the actin gene is now included in supplementary file 3. In addition, melting curves of the actin gene along 38 glucosinolate biosynthesis-related genes (revised) are presented in supplementary file 4.
Comment-2: Regarding the reference genes I would like to ask authors to selected the most appropriate pair of primers, from the three actin genes, to perform normalization using the geNorm software. I am not sure about the stability of Actin 3.
Responses: Based on your suggestions we have selected actin1. See cq values of actin1 in supplementary file 3.
Comment-3: Primers efficiency was included in Table S1 but R2 is still missing. The information included allow us to see efficiencies below 90%, the worst cases are Bol028919 and Bol032711 with the lowest efficiency (85%). In my point of view it would be required the design new primer pairs for all cases that showed efficiency below 90%. The information regarding the Bol028918 gene is missing.
Responses: We are sorry to miss out to include the efficiency of Bol028918 gene that will be 97% (see Table S1).
We have designed nine new primers to conduct expression analysis with desired amplification efficiencies ranging from 90% to 110% (see Table S1).
Comment-4: In my point of view there are results of gene expression analysis that remain not acceptable. The variability that I previously reported was not checked properly. The authors evaluate the inespecificity in an agarose gel and the observation of a single band was suficient to them to conclude that primers were specific. However, primers could amplify different fragment with very similar sizes but differences in the sequence, which could be related with the variability observed. The worst cases are the genes FMOGS-OX5-Bol031350 and CYP81F2-Bol026044, which in my point of view are not acceptable.
Responses: We repeated expression analysis with FMOGS-OX5-Bol031350 and CYP81F2-Bol026044 genes and obtained better melting peaks now.
Comment-5: In page 4: The PCA analysis remains for me very strange. I cannot understand why there are timepoints and glucosinolate components mixed, and also the reason why there are more samples in red than in blue.
Responses: There are five blue circles for susceptible line and five red rectangles for R line as we have five treatment × timepoint combinations (Control, mock at day1, mock at day 3, treated at day 1 and treated at day 3). We have drawn all figures and conducted ANOVA for those five interactions.
Comment-6: Line 343: “Since the nature of the signalling pathway could depends on the fungus type and the S. sclerotiorum is a necrotrophic fungus therefore it is likely that jasmonic acid (JA), ethylene (ET), and salicylic acid (SA), signalling pathway might be involved in resistance response of cabbage lines [62,63].” In my point of view this topic could be much better discussed. The authors are presenting the three signaling pathways, but it is known that one of those is more involved in plant response upon necrotrophic fungus. It will be interesting to have that information here presented and discussed.
Responses: Revised accordingly
Experimental evidence suggest that necrotrophic pathogens could be an alternative to jasmonic/ethylene (JA/ET) signaling pathway [63, 64] [Yang et al. 2010; Sticher et al. 1997] and that resistance against necrotrophs could be modulated by JA/ET signaling pathway [65,66] [Durrant and Dong 2004, Grant and Lamb 2006]. Since S. sclerotiorum is a necrotrophic fungus therefore it is likely that either JA or ET signalling pathway might be involved in resistance response of cabbage lines [67,68].
Comment-7: Line 398: “inoculation chamber” change it to growth chamber.
Responses: Thanks for your suggestion. Changed accordingly.
Comment-8: Regarding the sampling for gene expression analysis and metabolite analysis, it must be clear in the text that the sampling was made simultaneously.
Responses: Relevant sentence modified accordingly.
Leaf samples at 1 and 3 DPI were sampled simultaneously from the inoculated plants of the R line and the S line from each of the control, mock-treated, and S. sclerotiorum–infected plants to evaluate both the levels of endogenous GSLs and to quantify the expression of GSL biosynthetic pathway genes (Figure S1).
Comment-9: Line 409: Rephrase the sentence “Three biological replicates from three different plants of each of 45 inbred lines were inoculated into the third-youngest leaf of ninth-leaf-stage plants.” to “Three different plants of each of the 45 inbred lines (biological replicates) were inoculated into the third-youngest leaf of ninth-leaf-stage plants”.
Responses: Thanks for your advice. We inserted the revised sentence in manuscript.
Comment-10: Line 436: “from three randomly selected plants of the R line and the S line” It remaind unclear the procedure followed. If only three plants were inoculated how the selection was made ramdomly?
Responses: “from three randomly selected plants” has been replaced by “from the inoculated plants”.
Thanks for noticing this point.
Comment-11: Line 475: “from one B. oleracea actin genes (Supplementary file 5).” The information regarding the gene evaluated is not given and also the PCR conditions not presented in materials and methods section. Additionally it seems to be present a larger fragment arong 500pb (?), which could be related with inespecificity of primers, also seen in melting curve of Actin3.
Responses: Now PCR conditions inserted in materials and methods section.
RT-PCR was conducted using 1 μl of cDNA, 1 μl of forward and reverse primers (10 pmol concentration), 8 μl of Prime Taq Premix (2×) containing 1 U Taq polymerase (GENETBIO Inc., Korea) and 9 μl of double distilled water with a total volume of 20 µl. PCR condition: 5 min initial denaturation followed by denaturation at 94 °C and 25 cycles of denaturation at 94 °C for 30 sec, annealing at 56 °C for 30 sec, extension at 72 °C for 45 sec and final extension at 72 °C for 7 min. PCR products were ran on a 1.5% agarose gel along with a 100 bp size DNA ladder, stained with HiQ blue mango (20,000x), and pictured under UV light to obtain expected amplicons.
Revised:
from B. oleracea actin1 gene [58].
We have designed nine new primers to conduct expression analysis (see Table S1).
Comment-12: Line 479: primers concentration was not included and remains the volume added to PCR mix.
Responses: We used 1 µl of each forward and reverse primers (10 pmol) in 20 µl PCR mix.
Comment-13: The information regarding the evaluation of RNA quality is still not included.
Responses:
Purity of the extracted RNA was determined by the 260/280 nm ratio as quantified with a Nanodrop® ND-1000 and NanoDrop v3.7 software (Thermo Fisher Scientific, Waltham, MA, USA) (Supplementary file 5). The integrity of RNA was checked by agarose gel electrophoresis [77] (Laila et al. 2017).
Comment-14: The suggested changes for section 4.6 were not all considered. The information regarding primers specificity was not here included.
Responses: Information regarding the primers specificity has now been included.
In order to test the efficiency of primers, pooled cDNA of cabbage inbred lines of the same concentration of 300 ng·µL−1 was serially diluted 10 times per dilution. One µL diluted cDNA at 100, 10−1, 10−2, 10−3, 10−4 and 10−5 concentrations of original were used as templates in each reaction with forward and reverse primers. The qRT-PCR reaction was conducted for 40 cycles with melting curves in triplicate without template control. Semi-log plots of Ct versus fold dilutions were drawn to determine the slope. Primer efficiencies were calculated using the following equation: e = 10^(−1/slope). A value of slope of -3.321928 calculated 100% efficiency of a primer. Primers with efficiency levels of 90% to 100% were selected for expression analysis. Often primers were redesigned to obtain efficiency within the expected range.
Comment-15: In Line 465: “their efficiencies were tested” rewrite the sentence. Consider that efficiencies were not tested but calculated using parameters achieved by standard curve.
Responses: Revised. Thanks
Comment-16: At Supplementary file 5: RNA purity (260/280) is much lower than 2.0 which could lead to variability at the final results, not only do to RNA degradation (was this evaluated?) but also do to the presence of inhibitors at the solution.
Responses:
We rechecked our samples and found better RNA purity (see Supplementary file 5).
Thanks for your scholastic review. Regards and thanks,
Professor Ill-Sup Nou
Corresponding Author
Department of Horticulture
Sunchon National University, South Korea
Email: nis@sunchon.ac.kr
Round 3
Reviewer 2 Report
ijms-385957-peer-review-v2
Full Title: Glucosinolate profiling and expression analysis of Glucosinolate biosynthesis genes differentiate white mold resistant and susceptible cabbage lines
General comments to the manuscript:
In my point of view the manuscript is now acceptable for publication. Nevertheless, I have few small points that must be considered to be improved.
Line 161: Regarding the text that authors included “The forward and reverse primer of actin1 was designed from the sequence obtained from the NCBI database (GenBankAccession no. AF044573) [58] since that gene was expressed in both the R and S lines and was used as a reference.” I would suggest to rewrite it like I am suggesting below:
“The forward and reverse primers of Actin1 were designed based on the sequence available at NCBI database (GenBankAccession no. AF044573). The selection of that gene as reference gene to perform expression data normalization was based on previous reports showing the stability of that gene on the same plant species upon similar experimental conditions [35,58].”
Line 344: I would suggest to change the sentence ” Experimental evidence suggest that necrotrophic pathogens could be an alternative to jasmonic/ethylene (JA/ET) signaling pathway [63,64] and that resistance against necrotrophs could be modulated by JA/ET signaling pathway [65,66].” to “Experimental evidences suggested that resistance against necrotrophic pathogens could be modulated by JA/ET signaling pathway [65,66].”
Line 440: I would suggest to change the sentence “Leaf samples at 1 and 3 DPI were sampled simultaneously from the inoculated plants of the R line…” to “Leaf samples at 1 and 3 DPI were simultaneously sampled from plants of the R line…”
Author Response
Response to Reviewer 2- Round 3
ijms-385957-peer-review-v2
Full Title: Glucosinolate profiling and expression analysis of Glucosinolate biosynthesis genes differentiate white mold resistant and susceptible cabbage lines
Comments:
In my point of view the manuscript is now acceptable for publication. Nevertheless, I have few small points that must be considered to be improved.
Response: Thanks for accepting our manuscript.
Line 161: Regarding the text that authors included “The forward and reverse primer of actin1 was designed from the sequence obtained from the NCBI database (GenBankAccession no. AF044573) [58] since that gene was expressed in both the R and S lines and was used as a reference.” I would suggest to rewrite it like I am suggesting below:
“The forward and reverse primers of Actin1 were designed based on the sequence available at NCBI database (GenBankAccession no. AF044573). The selection of that gene as reference gene to perform expression data normalization was based on previous reports showing the stability of that gene on the same plant species upon similar experimental conditions [35,58].”
Response: Revised accordingly. Thanks
Line 344: I would suggest to change the sentence “Experimental evidence suggest that necrotrophic pathogens could be an alternative to jasmonic/ethylene (JA/ET) signaling pathway [63,64] and that resistance against necrotrophs could be modulated by JA/ET signaling pathway [65,66].” to “Experimental evidences suggested that resistance against necrotrophic pathogens could be modulated by JA/ET signaling pathway [65,66].”
Response: Revised accordingly. Thanks
Line 440: I would suggest to change the sentence “Leaf samples at 1 and 3 DPI were sampled simultaneously from the inoculated plants of the R line…” to “Leaf samples at 1 and 3 DPI were simultaneously sampled from plants of the R line…”
Response: Revised accordingly. Thanks
Thanks for your scholastic review. Regards and thanks,
Professor Ill-Sup Nou
Corresponding Author
Department of Horticulture
Sunchon National University, South Korea
Email: nis@sunchon.ac.kr